# Research on Economic Optimal Dispatching of Microgrid Based on an Improved Bacteria Foraging Optimization

**DOI:** 10.3390/biomimetics8020150

**Published:** 2023-04-07

**Authors:** Yi Zhang, Yang Lv, Yangkun Zhou

**Affiliations:** College of Electrical and Computer Science, Jilin Jianzhu University, Changchun 130000, China

**Keywords:** bacterial foraging optimization, microgrid, distributed generation, energy consumption, renewable energy

## Abstract

This paper proposes an improved Bacterial Foraging Optimization for economically optimal dispatching of the microgrid. Three optimized steps are presented to solve the slow convergence, poor precision, and low efficiency of traditional Bacterial Foraging Optimization. First, the self-adaptive step size equation in the chemotaxis process is present, and the particle swarm velocity equation is used to improve the convergence speed and precision of the algorithm. Second, the crisscross algorithm is used to enrich the replication population and improve the global search performance of the algorithm in the replication process. Finally, the dynamic probability and sine-cosine algorithm are used to solve the problem of easy loss of high-quality individuals in dispersal. Quantitative analysis and experiments demonstrated the superiority of the algorithm in the benchmark function. In addition, this study built a multi-objective microgrid dynamic economic dispatch model and dealt with the uncertainty of wind and solar using the Monte Carlo method in the model. Experiments show that this model can effectively reduce the operating cost of the microgrid, improve economic benefits, and reduce environmental pollution. The economic cost is reduced by 3.79% compared to the widely used PSO, and the economic cost is reduced by 5.23% compared to the traditional BFO.

## 1. Introduction

The electricity demand continues to expand with social science and technology development. It is challenging to meet electricity demand using only traditional thermal power generation methods. Distributed generation (DG) is widely used in microgrid systems and supplies power to regional users because of its flexible control, self-protection, and schedulability. The energy consumption problems caused by traditional power generation methods can be alleviated, and the environmental pollution caused by thermal power generation can be reduced. Wind and photovoltaics are clean and renewable energy sources in microgrids. However, the intermittence and fluctuations caused by wind speed and light intensity will challenge the stable operation of the grid system with a large-scale grid connection of wind turbines and photovoltaic panels. Therefore, it is necessary to first deal with the uncertainty of wind photovoltaics and then use DG, which can operate flexibly and cooperate with renewable energy generation technologies to build a microgrid and optimize its scheduling. This model effectively reduces economic costs and improves power supply quality and stability.

Many scholars have studied the uncertainty treatment of wind and light. Ref. [1] proposed a Latin hypercube sampling method to process the uncertainty of wind and solar data, effectively reducing the impact of wind and light on power grid peak shaving [2]. Combining the roulette wheel mechanism and Monte Carlo thinking to process wind and light data, using randomly generated scenes to simulate the uncertain process of scenery, this method can effectively reduce the economic cost of hydrothermal unit scheduling.

Linear programming [3,4,5], dynamic programming [6,7], Lagrangian relaxation [8], and nonlinear programming [9,10] have many problems with large-scale power systems, such as traditional microgrid dispatching optimization methods. Various electrical constraints also increase the complexity and difficulty of microgrid dispatching optimization. These problems mainly focus on accuracy and computational efficiency. An increasing number of swarm intelligence algorithms have been applied to microgrid dispatching optimization to overcome such problems. Some mature swarm intelligence algorithms, such as PSO [11], GA [12], and WOA [13] are widely used in microgrid dispatching optimization due to their advantages of fast convergence and simple processes. However, these traditional algorithms also easily fall into local optima, have low efficiency, premature convergence results, and low precision. Many optimization methods have been encountered in the process of microgrid dispatching optimization. Reference [14] introduced a simulated annealing algorithm and chaos optimization into the PSO algorithm, enriching the population diversity, and enhancing the global search ability. Reference [15] presented a differential evolution algorithm (ED) into the quantum particle swarm optimization algorithm (QPSO), which improved the ability to jump out of the optimal local solution in the later stage. Reference [16] proposed using tabu search to optimize particle swarm optimization compared with CLQPSO and short-term microgrid scheduling results. The IPSO is better than the traditional PSO and Tabu Search (TS) in the two-level energy optimization scheduling strategy. Reference [17] proposed a BPSO method for scheduling household energy management systems with distributed power sources. This method can effectively reduce economic costs, energy consumption, and environmental pollution. Reference [18] used an adaptive strategy to optimize the GA to improve its convergence accuracy of the GA algorithm.

Moreover, Reference [19] combined the advantages of the GA and PSO algorithms. The microgrid dispatching optimization was modeled as a quadratic programming problem, and the improved GA-PSO algorithm was used. The WOA has a higher solution quality than the PSO and GA. However, WOA also needs help with problems, such as premature convergence and low accuracy of the results. To improve the performance of WOA, Reference [20] uses adaptive inertia weight, spiral search method, and generalized inverse learning to improve and optimize WOA. The results of multiregion interconnected microgrid system scheduling prove that this optimization method can effectively improve the performance of the WOA and reduce the cost of microgrid operation. New swarm intelligence algorithms have been gradually developed and applied to microgrid economic dispatch models. Reference [21] proposed an economic dispatch model using the cuckoo algorithm to optimize multiple microgrids and determined the power supply strategy by predicting the best state of charge of the battery. This method has a significant improvement over PSO. Reference [22] utilized a mixture of bacterial foraging algorithms and genetic algorithms to achieve minimum cost load management [23]. The bacterial foraging algorithm is also used to achieve the scheduling of isolated microgrids, and a large number of experiments have proven the effectiveness of the proposed method, which can reduce economic cost. Reference [24] proposed a bald eagle search optimization algorithm (BESOA) to control the scheduling between demand and power supply, which can effectively reduce energy costs and microgrid emissions costs. Reference [25] proposed applying the butterfly algorithm to the microgrid scheduling solution. Aiming at the problem of poor convergence accuracy of the butterfly algorithm and easy falling into a local optimum, they used Cauchy mutation to improve the position information of the butterfly and expand the global search performance of the algorithm by using chaotic mapping enriched species diversity. The method’s effectiveness is proven by scheduling a microgrid cluster system composed of multiple microgrids. An improved crow algorithm [26] is proposed for microgrid scheduling with distributed power sources. By introducing a Levy flight strategy, the convergence speed and result accuracy of the algorithm can be effectively improved.

Most swarm intelligence research focuses on solving the performance of algorithms in the microgrid dispatching process. (2) The main contributions of this study are as follows:
(1)This study improved the algorithm’s speed and considered its accuracy in chemotaxis. The adaptive step size formula replaces the standard fixed step size, and the PSO speed formula is introduced to improve the random direction vector (PHI).(2)The crisscross algorithm is used to improve the population of the algorithm and global search performance in the replication part.(3)The dynamic dispersal equation and sine-cosine algorithm were used to improve the loss of high-quality results and the algorithm’s efficiency for the dispersal part.

The remainder of this paper is organized as follows: the first section builds a microgrid model with multiple objective cost functions; the second section describes the improvement of the BFO, quantitatively analyzes the impact of each part of the improvement, and conducts a comparative test; the third section deals with the uncertain processes of wind and solar, applies the BFO to the microgrid dispatching model to solve it, and gives the comparative experimental results; and the fourth section is a summary of the full text.

## 2. Microgrid Economic Dispatch Model

The microgrid example comprises distributed power sources, such as wind generators, photovoltaic power generation panels, diesel engines, micro steam turbines, and fuel cells shown in Figure 1.

### 2.1. The Model of DG

As energy generation forms, wind and photovoltaic power have significant volatility and are affected by many meteorological factors, such as wind speed, wind direction, light intensity, and temperature. Therefore, the power balance brought by wind power generation has great uncertainty. A model of a wind turbine follows:(1)$$P_{W}=\{\begin{array}{c} 0,v<v_{in}\;or\;v<v_{out}\\ P_{n},v_{n}\leq v\leq v_{out}\\ \frac{\left(v-v_{in}\right)}{\left(v_{n}-v_{in}\right)}P_{n},v_{in}\leq v\leq v_{n} \end{array}.$$

In Equation (1), (Explain the specific meaning of the formula) $P_{W}$ represents the output of wind power, $P_{n}$ represents rated power, and the output of wind power generation is related to the actual wind speed, where v represents actual wind speed, $v_{in}$, $v_{out}$ and $v_{n}$ represent the cut-in wind speed, cut-out wind speed, and rated wind speed, respectively. When the wind speed is less than the cut-in wind speed or greater than the cut-out wind speed, the fan is not working. The output power is zero. If the wind speed is between the rated wind speed and the cut-out wind speed, the fan output is at the rated power output. When the wind speed is between, the output power of the fan is reflected by the empirical equation.

Photovoltaic panel model:(2)$$P_{V}=rs\eta_{V}.$$

Photovoltaic output is positively correlated with light intensity. In Equation (2), $P_{V}$ represents the output power of solar energy, r represents the light intensity, s represents the total radiation area of the photovoltaic module, and $\eta_{V}$ represents the photoelectric conversion efficiency of the photovoltaic panel.

Diesel generator model:

The mathematical model of diesel generators is like that of coal-fired units in thermal power generation, and it has a particular peak-shaving effect in microgrids. The formula is as follows:(3)$$C_{DE}=\sum_{i=1}^{N}\left(a_{i}+b_{i}P_{DE}+c_{i}P_{DE}^{2}\right).$$

In Equation (3), $a_{i}$, $b_{i}$, and $c_{i}$ are the cost coefficients of diesel generators; $C_{DE}$ represents the cost of power generation; $P_{DE}$ represents the output of generators; *N* represents the number of diesel generators.

Micro Turbine Model:

The stand-alone power of a microturbine is small, generally between 20 and 300 KW, and its formula is as follows:(4)$$C_{MG}=\frac{Price}{L}\sum \frac{P_{MG}\left(t\right)\Delta t}{\eta_{MG}\left(t\right)}.$$

In Equation (4), *Price* and *L* represent the price and calorific value of natural gas, respectively; $P_{MG}\left(t\right)$ represents the output power of the microturbine at time *t*; $\eta_{MG}\left(t\right)$ represents the power generation efficiency at this time, and the efficiencies of different types of micro-turbines can be obtained by polynomial curve fitting. The model of the micro turbine is C65; Δt indicates the operating hours.

Fuel cell model:

Fuel cells are known as the fourth-generation power generation device technology. Proton exchange membrane fuel cells are selected, and their cost formula is similar to that of microturbines:(5)$$C_{FC}=\frac{Price}{L}\sum \frac{P_{FC}\left(t\right)\Delta t}{\eta_{FC}\left(t\right)}.$$

In Equations (5), $P_{FC}\left(t\right)$ and $\eta_{FC}$ represent the fuel cell’s output power and power generation efficiency, respectively, where the power generation efficiency can also be obtained by polynomial curve fitting.

The traditional neural network prediction method for wind and solar output prediction is unsuitable because wind and solar have volatility and randomness and are affected by many factors. Traditional wind and solar output forecasting use a neural network to make regression predictions on historical power. However, the predicted value often needs to reflect the influence of uncertain weather conditions on wind and solar output. Aiming at the uncertainty processing of wind and solar, Ref. [27] proposed a two-stage optimization model of the microgrid from the two stages of day ahead and real time. This method can effectively reduce the prediction error and improve the stability of the power supply. Ref. [28] proposed a microgrid interval optimization method based on affine arithmetic and used the non-dominated sorting genetic algorithm to solve the framework. The results show that this method can effectively alleviate the uncertainty brought by renewable energy, such as wind and solar. Ref. [29] considered the meteorological conditions comprehensively, using a data-driven Bayesian non-parametric method, modified column, and constraint generation (CC) to solve the uncertainty problem of the scenery further. This paper uses the Monte Carlo sampling method to reduce the scene of wind and solar generation. The uncertain process of wind power and photovoltaic output is simulated by randomly optimized scenario generation and reduction. The core idea of Monte Carlo is to use the frequency of events to approximate the probability of event occurrence and use the Euclidean distance as the basis for scene reduction to iterate. The method is mainly divided into three parts: building a probability model, sampling, and valuation.

Weibull distribution of wind speed:(6)$$f\left(v,c,k\right)=\left(\frac{k}{c}\right){\left(\frac{v}{c}\right)}^{k-1}e^{-{\left(\frac{v}{c}\right)}^{k}}.$$

In Equation (6), *v* represents the actual wind speed, and *c* and *k* are Weibull’s model parameters.

Beta distribution of light intensity:(7)$$f\left(s,\alpha ,\beta \right)=\frac{\Gamma \left(\alpha +\beta \right)}{\Gamma \left(\alpha \right)\Gamma \left(\beta \right)}s^{\left(\alpha -1\right)}{\left(1-s\right)}^{\left(\beta -1\right)}.$$

In Equation (7), *s* represents the light intensity α and β is the shape parameter of the Beta distribution. The wind speed and light intensity data fitted by Weibull and Beta distributions are, respectively, substituted into the wind turbine output model and the photovoltaic power generation output model. The wind and solar output prediction is obtained by using Monte Carlo scene reduction.

### 2.2. Microgrid Economic Dispatching Model

There are multiple objective functions and constraints in the microgrid dispatching optimization problem, and most of the research on microgrid scheduling only considers the cost target of the DG output of the microgrid and the cost of electricity purchase and sale. In this paper’s microgrid economic dispatching model, there are traditional power generation, such as microturbines and diesel generators, so the three objective functions of DG cost, environmental impact cost, and power purchase cost are considered comprehensively.

Power Generation Cost:(8)$$F_{1}=\min\left(\sum_{t=1}^{T}\left[\sum_{i=1}^{N}\left(\left(C_{i,f}+C_{i,m}\right)P_{i,t}\right)\right]\right).$$

In Equation (8), *T* represents the scheduling cycle; *N* represents the total number of distributed power sources; $C_{i,f}$ and $C_{i,m}$￼, respectively, $P_{i,t}$ cost and equipment maintenance cost of distributed power sources; ￼ represents the output result of the *i*-th power source at time *t*. The installation and maintenance costs of wind turbines and photovoltaic power generation panels are not considered here.

Environmental Impact Costs:

In the microgrid of this example, there is a distributed power source that uses natural gas, diesel, and other energy supplies as raw materials for power generation. It is also necessary to include the environmental impact cost in the objective function of microgrid dispatching optimization to consider the environmental gas pollution caused by the consumption of these raw materials (mainly refers to $CO_{2}$, $SO_{2}$, $NO_{X}$, etc.):(9)$$F_{2}=\min\left(\sum_{t=1}^{T}\left(\sum_{i=1}^{N}\left(C_{i,e}P_{i,t}\right)\right)\right).$$

Equation (9), $C_{i,e}$ represents the pollution gas environmental cost coefficient corresponding to each distributed power supply.

Electricity price cost:

To reflect intuitively the consumption and utilization of the microgrid, consider the cost of electricity price as the objective optimization function:(10)$$F_{3}=\min\left(\sum_{t=1}^{T}\left(C_{grid,t}P_{grid,t}\right)\right).$$

In Equation (10), $C_{grid,t}$ represents the real-time electricity price (24 h system); $P_{grid,t}$ represents the power exchange result between the microgrid and the enormous power grid after considering the user load.

In summary, the objective function of microgrid dispatching optimization can be expressed as
(11)$$\min F=\sum_{i=1}^{3}F_{i}.$$

Micro Turbine Constraints:(12)$$P_{MG}^{\min}\leq P_{MG}\left(t\right)\leq P_{MG}^{\max}.$$

In Equation (12), $P_{MG}^{\min}$ represents the minimum output constraint of the microturbine; $P_{MG}^{\max}$ represents the maximum power of the microturbine unit output; some studies use the rated power here.

Diesel Constraints:(13)$$P_{DE}^{\min}\leq P_{DE}\left(t\right)\leq P_{DE}^{\max}.$$

Fuel Cell Constraints:(14)$$P_{FC}^{\min}\leq P_{FC}\left(t\right)\leq P_{FC}^{\max}.$$

Power exchange constraints between the microgrid and main grid:(15)$$P_{grid}^{\min}\leq P_{grid}\leq P_{grid}^{\max}.$$

In the above equation, $P_{grid}^{\min}$$P_{grid}^{\max}$ ￼ represent the minimum and maximum power allowed for power exchange between the microgrid and the primary grid.

Microgrid supply and demand balance constraints:(16)$$P_{L,t}=P_{grid,t}+P_{MG,t}+P_{DE,t}+P_{FC,t}+P_{W,t}+P_{V,t}.$$

In Equation (16), $P_{L,t}$ represents the total power demanded by the load during the t period.

## 3. An Improved Bacterial Foraging Optimization and Its Application

The BFO [30,31,32] is a new swarm intelligence algorithm mainly divided into chemotaxis, replication, and dispersal. (BFO’s Short Insights) Traditional BFOs will face problems, such as slow convergence speed, poor accuracy of results, easy to fall into local optimization, and low efficiency of algorithms. The schematic diagram of *E. coli* is shown in Figure 2, Figure 2a represents the forward swimming process of *E. coli*, and Figure 2b represents the overturning process of *E. coli*. 

### 3.1. Chemotaxis Process

Chemotaxis is a significant part of the BFO algorithm. It simulates the forward and reverse of *Escherichia coli* in the foraging process, as shown in the above picture (Figure 2). This is used as the primary means of optimization. The following formula can approximate the process of flipping forward:(17)$$\begin{array}{l} \theta \left(i,j+1,k,l\right)=\theta \left(i,j,k,l\right)+C\left(i\right)PHI\\ PHI=\frac{\Delta i}{\sqrt{\Delta^{T}i\cdot \Delta i}} \end{array}$$

In Equation (17), θ(i,j,k,l) represents the position of the *i*-th bacterium at the *j*-th chemotaxis, *k*-th replication, and *l*-th dispersal; C(i) represents the step size of the *i*-th bacterium, and the traditional Bacterial Foraging Optimization adopts a fixed step size formula to find the Optimal solution; PHI represents the random direction of bacteria forward; Δi is the defined random direction vector.

Step size *C* has a significant impact on the convergence speed and accuracy of the algorithm. Although a more significant step size can increase the convergence speed, it reduces the accuracy. Although a too-small step size can improve the solution accuracy, it will cause the algorithm to converge slowly. The fixed step size in the traditional BFO is a fundamental reason for the slow convergence of the algorithm because it cannot balance the convergence speed and accuracy. Ref. [33] introduced the step size search formula of the fish swarm algorithm into the BFO and used an adaptive function to improve the step size. The speed of this function is very slow at the beginning of the iteration, and the speed suddenly increases at the end of the iteration. Although this method can improve the convergence speed to a certain extent, its impact on the global search performance of the algorithm remains to be verified. For this reason, this paper proposes to use the dynamic step size formula instead of the traditional fixed value.
(18)$$C\left(x\right)=\exp\left(-{\left(N_{c}N_{re}N_{ed}-\frac{\tau }{j+\left(k-1\right)N_{c}+\left(l-1\right)N_{re}N_{ed}}\right)}^{\frac{1}{\alpha }}\right)C.$$

In Equation (18), (We rewrote the correlation equation) $N_{c}$, $N_{re}$, and $N_{ed}$, respectively, denote the number of chemotaxis restriction, replication restriction, and dispersal restriction in the α￼ expressed as the step length coefficient. The above equation makes it possible to search with a more significant step size in the early stage of the algorithm iteration and speed up the algorithm’s convergence speed. As the number of iterations increases, a refined search is performed with a small step size in the later stage of the iteration to improve the algorithm’s accuracy.

Another factor that plays a crucial role in the convergence speed is the random direction vector PHI. Due to the random process being included in the definition of the random direction vector, the convergence speed of the algorithm is limited [34]. For this reason, this paper proposes a particle swarm algorithm speed formula using dynamic inertia weight factors to replace the traditional random direction vector.
(19)$$V\left(j+1\right)=w\_nowV\left(j\right)+c_{1}r_{1}\left(Pbest_{i}-P_{i}\right)+c_{2}r_{2}\left(Gbest_{i}-P_{i}\right),$$
(20)$$w\_now=\frac{\left(w\_start-w\_end\right)\left(Maxiter-iter\right)}{Maxiter}+w\_end.$$

In Equation (19), *V* represents the velocity; $c_{1}$ and $c_{2}$ represent the weight coefficient; $r_{1}$ and $r_{2}$ represent random number; $Pbest_{i}$$Gbest_{i}$￼ represent the current optimal solution and the optimal global solution, respectively w_now￼ represents the dynamic inertia weight coefficient. The advantages of the fast solution speed of the particle swarm optimization algorithm are mainly reflected in the speed and population update formula. The update of velocity *V* depends on the optimal solution, which is very different from random optimization. Many invalid searches are avoided, which is also the main reason for the fast iteration of the particle swarm optimization algorithm. Replacing the traditional PHI with the speed formula can speed up the chemotaxis speed of bacteria foraging and significantly improve the convergence speed and efficiency of the algorithm (Algorithm 1).
**Algorithm 1:** Chemotaxis process with hybrid dynamic step size and PSO *1 for j = 1:Nc**2  for i = 1:s**3    C = C(x) (C (x) represents the dynamic adaptive step size of the xth bacteria)**4    Calculate the influence of bacterial clustering behavior on fitness value and save as Jl**5    Replacing PHI with Particle Swarm Velocity Formula**6    P(i,j + 1) = P(j) + C * PHI**7    Update fitness value J**8    while (m < Ns)**9      if (J < Jl)**10       Update fitness value J**11      else**12       m = Ns**13      end**14   end**15   Update fitness value Jl**16  end**17 end*

### 3.2. Replication Process

The replication process is the process of simulating the biological elimination competition of bacteria. After the chemotaxis is completed, the replication operation is performed according to the accumulated health value of the bacteria, namely:(21)$$J_{Health}^{i}=\sum_{j=1}^{N_{c}}J\left(i,j,k,l\right).$$

In Equation (21), $J_{Health}^{i}$ represents the cumulative health value of bacteria *i*; $N_{c}$ which represents the total number of chemotaxis. The BFO algorithm uses the binary replication method. The cumulative health value is sorted in ascending order, and the first half of high-quality bacterial individuals are copied to keep the overall number of bacteria unchanged.

The traditional binary replication method has certain disadvantages. Although the algorithm’s complexity is reduced, the diversity of the population is also significantly reduced. Currently, the improvement of the BFO replication process mainly focuses on the hybridization of the population, and the commonly used methods include mixed GA. Ref. [35] proposed the method of crossing bacterial individuals to improve the population. However, this paper proposes to use the crisscross algorithm [36,37,38] to replace the binary replication method to improve the replication process and ensure the diversity of the population. The crisscross algorithm has been an emerging swarm intelligence algorithm in recent years. Each crossover iteration of the crisscross has a comparison process with the previous generation different from Ref. [35]. The horizontal crossover process is as follows:(22)$$MS_{hc}\left(x,d\right)=r_{1}X\left(x,d\right)+\left(1-r_{1}\right)X\left(y,d\right)+c_{1}\left(X\left(x,d\right)-X\left(y,d\right)\right),$$
(23)$$MS_{hc}\left(y,d\right)=r_{1}X\left(y,d\right)+\left(1-r_{1}\right)X\left(x,d\right)+c_{1}\left(X\left(y,d\right)-X\left(x,d\right)\right).$$

This equation is expressed as the intersection of X(x,d) and X(y,d) in the d dimension, which $r_{1}$ represents a random number between [0, 1] and $c_{1}$ represents a random number between [−1, 1]. Two individuals can be crossed to produce two offspring, which can be decided by updating the health value. High-quality individuals enter the next step of the vertical crossover process.

The vertical cross process is:(24)$$MS_{vc}\left(x,d_{1}\right)=rX\left(x,d_{1}\right)+\left(1-r\right)X\left(x,d_{2}\right).$$

This formula is expressed as crossing bacteria *x* in two dimensions. The vertical crossing is a random process. The vertical crossing will be performed only when the satisfaction probability is less than the crossing probability. This is also the difference from the horizontal crossing. At the same time, it is only better than the parent generation. The cross children will be retained for the next iteration. Relying on the CSO to update the population in different dimensions can effectively broaden the population’s diversity and enhance the algorithm’s global search performance (Algorithm 2).
**Algorithm 2:** Replication Process of Hybrid CSO *1 for k = 1:Nre**2   for i = 0:s/2 − 1**3    if rand < Longitudinal crossing probability**4     for j = 1:p**5       Longitudinal crossing of populations**6     end  **7    end**8   end**9   Update population according to fitness value**10  for i = 0:p/2 − 1**11    for j = 1:s**12     Horizontal crossing of populations**13    end**14  end**15 end*

### 3.3. Dispersal Process

Dispersion is significant to ensure that the algorithm jumps out of the optimal local solution. The algorithm is designed to regenerate a bacterium *i* according to the initial population generation formula if the random probability r is less than the fixed dispersal probability $P_{ed}$. This method simulates the influence of the external environment on *E. coli*. The traditional fixed dispersal probability will bring certain disadvantages. Some high-quality bacterial individuals will be eliminated, thereby reducing the algorithm’s efficiency under the fixed probability to satisfy the algorithm’s general nature. This paper introduces a dynamic probability formula to replace the traditional fixed dispersal probability through:(25)$$P\left(x\right)=P_{ed}\frac{J_{worst}-J_{x}}{J_{worst}-J_{best}}.$$

The probability can be adjusted to a dynamic probability that changes with the health value. In Equations (25), $J_{worst}$ represents the worst value of the health degree, $J_{best}$ represents the optimal value of the health degree, and $J_{x}$ represents the real-time health value of the *x*-th bacteria. The probability of individuals with excellent health values being dispersed decreases while the probability of bacterial individuals with poor health values being dispersed increases in this way. Thus, the high-quality individuals avoid loss and ensure the efficiency and performance of the algorithm. Although, the dynamic dispersal probability can avoid the loss of high-quality solutions as much as possible. This paper proposes the Sine-Cosine algorithm (SCA) to improve the dispersal process.

SCA is a new swarm intelligence algorithm proposed by Australian scholar Mirjalili in 2016 [39]. The algorithm utilizes sine and cosine functions inspired by the fluctuating optimization of sine and cosine functions to fluctuate the initial random candidate solution toward the optimal solution or vice versa to complete the optimization process.
(26)$$X_{i}^{t+1}=X_{i}^{t}+r_{1}\sin\left(r_{2}\right)\left|r_{3}P_{i}^{t}-X_{i}^{t}\right|,r_{4}<0.5,$$
(27)$$X_{i}^{t+1}=X_{i}^{t}+r_{1}\cos\left(r_{2}\right)\left|r_{3}P_{i}^{t}-X_{i}^{t}\right|,r_{4}\geq 0.5.$$

The above equation $X_{i}^{t}$ represents the position of the *i*-th dimension of the current solution in the *t*-th iteration, the value of $r_{1}$ to $r_{4}$ means a random number, indicating the end position of the *i*-th dimension. $r_{4}$ is a random number that represents the determination of the probability of a search strategy. Taking 0.5 allows the two strategies to be performed with equal probability. When the random number is less than 0.5, a sinusoidal oscillation search is performed, and when the random number is greater than 0.5, a cosine oscillation search is performed. Ref. [40] applies the SCA to the process of chemotaxis and uses the sine-cosine search formula to optimize the random direction vector PHI so that the step size can be reduced linearly from a to 0. The speed of BFO improves in this model. This paper proposes using the SCA to improve population generation in dispersal. After satisfying the dynamic dispersal probability, it is judged again to perform the sine-cosine search. The terminal position is set as the optimal bacterial position, and the above formula generates dispersed populations of bacteria that meet the requirements. The dispersed bacteria will change with the optimization process of the algorithm. Thus, the loss of the optimal value caused by the randomness of traditional methods is avoided, and the algorithm’s efficiency is improved (Algorithm 3).
**Algorithm 3:** Dispersion process of hybrid dynamic probability and SCA*1 for l = 1:Ned**2   for m = 1:s**3    Dynamic dispersion probability**4    if Ped > rand**5     if r4 < 0.5**6      Sinusoidal oscillation search**7     else**8      Cosine oscillation search**9     end**10    end**11  end**12 end*

### 3.4. Test Analysis

To quantitatively analyze the impact of each improved part on the performance of the algorithm, this paper uses six test functions to test BFO and BFO with CSO and BFO with SCA and BFO with CSO and SCA, BFO with PSO, IBFO with PSO, and CSO and SCA. Moreover, this paper compared BFO with PSO and some improved bacterial foraging algorithms. The number of iterations is 400, the number of bacteria s is 50, the number of chemotaxis *Nc* is 50, the maximum number of steps *Ns* of one-way movement is 4, the number of copy operations *Nre* is 4, the number of dispersal *Ned* is 2, the traditional dispersal probability *Ped* is 0.25, the number of attractants d attract is 0.1, the release rate of the attractant ommiga attract is 0.2, the number of repellants h repellant is 0.1, and the release rate ommiga repellant of the repellant is 0.1. Each group of test functions is carried out in 20 experiments, and the test results are as follows (Table 1, Figure 3, Figure 4, Figure 5, Figure 6, Figure 7 and Figure 8).

The ordinate in the figure above represents the function value, and the abscissa represents the iteration number. By combining the algorithm’s test results and convergence curve, SCA comprehensively improves the convergence speed and accuracy of the algorithm. The CSO improves the algorithm’s accuracy more significantly but at the expense of convergence speed. Mixing the two will improve the accuracy due to SCA’s efficient population dispersal during the dispersal process. It reduces the algorithm speed slightly simultaneously because mixing crisscross takes some time. However, the CSO enriches the replication population and improves the search performance of the algorithm. The convergence result will change abruptly due to the influence of dispersal on population initialization when iteration is 200 times. Improving the dispersal process can effectively alleviate this situation and make the whole process of algorithm convergence relatively smooth. Mixing PSO can significantly improve the convergence rate of BFO because PSO can improve the process of *E. coli* rollover swimming and avoid a lot of invalid random searches. The speed update formula of the particle swarm optimization algorithm can quickly search for iteration. The BFO integrated with various optimization parts has the highest search accuracy and the fastest iteration speed. This paper also compares some other algorithms for BFO optimization, such as the Hormone Regulation based Emotional Bacterial Foraging Algorithm (HR-EBFA) [41], Bacterial Foraging reinforcement Learning Optimization Algorithm (RL-BFA) [42], the improved Quantum Bacterial Foraging Algorithm (MQBFA) [44], and Distribution Estimation based adaptive Bacterial Foraging Algorithm (BFOED) [43]. The improved bacterial foraging algorithm presented in this paper has better results than the above methods.

## 4. Algorithm Application and Experimental Analysis

In this paper, microgrid dispatching optimization is designed as the behavior of bacteria foraging. The flow chart is as follows:

Among them, in Figure 9, the bacterial population is initialized as a six-dimensional array: $P=zeros\left(N,T,popsize,N_{c},N_{re},N_{ed}\right)$. The first to sixth dimensions correspond to the distributed power supply, the microgrid operating period, the number of bacteria, the number of chemotaxes, the number of replications, and the number of dispersals.

### 4.1. Uncertainty Treatment of Wind Power and Photovoltaic

Given the uncertain influence of wind power and solar, this paper uses meteorological data from a particular place in Inner Mongolia to predict wind and solar output. It uses the MC’s idea to generate 100 scenery scenes and then cut them back randomly. The five output scenes and their probabilities after reduction are shown in the figure below.

According to Figure 10, Scenario 5 has the highest probability. Figure 11 shows 100 random scenarios for simulating wind power output, and Figure 12 shows 100 random scenarios for simulating photovoltaic output. Figure 13 and Figure 14 show the output after scenario reduction. The uncertainty process of wind and photovoltaic output are simulated by using the probability of scenario occurrence. Therefore, this paper uses the forecast data of the reduced wind and photovoltaic output of Scenario 5, and the reduced wind power output is shown in Figure 15.

### 4.2. Examples of Microgrid Dispatching

The basic parameters of the controllable distributed power generation in Table 2 and typical daily load and real-time electricity price in Table 3 in this example are as follows:

The typical daily load [45] and real-time electricity price [46] are as follows:

Considering the environmental cost, the emission coefficients of each pollutant, and the corresponding cost coefficients [47] are as follows (Table 4):

Figure 16 shows the scheduling results of the microgrid. Wind and photovoltaic are clean and renewable energy that will be fully output in the microgrid dispatching, and other distributed power sources will fill the shortage of unmet load demand. It is necessary to purchase power from the enormous power grid if the combined output of each distributed power source under the constraint of the upper limit of the maximum power fails to meet the load demand. Fuel cells are prioritized for most of the operation period, followed by micro steam turbines and diesel generators because of their low operating costs and environmental impact. The overflow will be sold to the grid according to the selling price between hours three and six; the overflow occurs at hours nineteen and twenty-two. This study will consider adding energy storage equipment into the subsequent research to improve the absorption capacity of the power grid.

It can be seen from Figure 17 that the Total cost (T-cost) includes various parts of the cost, including the operation cost of distributed power supply (O-cost), Environmental pollution cost (E-cost), and the cost of electricity purchase and sale (P and S-cost). The operation cost of distributed power supply gradually increases after the start-up of each distributed power supply and then fluctuates within a specific range; during the stop phase, the operation cost will gradually decrease. O-cost accounts for a large proportion of T-cost, followed by E-cost and P-and S-cost. The negative value of the P- and S-cost represents the profit from electricity sales, and the positive value represents the electricity purchase expenditure.

The following Table 5 shows the detailed data of each distributed power source and power grid output in the optimal scheduling results of the microgrid.

This paper applies IBFO, BFO, and PSO to the microgrid economic dispatching problem to verify the improved method’s excellence. The number of experiments is 30, and the experimental results are averaged. In Table 5, the experimental result of IBFO is the best, which is CNY 1653.4, the experimental result of BFO is CNY 1739.9, and the experimental result of PSO is CNY 1716.0. The microgrid dispatching cost of IBFO is reduced by 3.79% compared with PSO. The scheduling cost of the improved bacterial foraging algorithm is reduced by 5.23% compared with the traditional BFO. The experimental results show that the improved bacterial foraging algorithm can effectively reduce the cost of economic dispatching optimization of the microgrid.

## 5. Conclusions

This paper refines the three main steps of BFO. This study introduced the speed formula of PSO in the chemotaxis stage to improve the search accuracy and speed. The adaptive step size is used to modify the standard fixed step size, which made the algorithm search at high speed in the early iteration. The improvements reduce the step size in the late iteration. The CrissCross Algorithm is used to enrich the population diversity in the replication stage. The global search performance of the algorithm is further enhanced. This research proposes the dynamic dispersal equation to improve the survival of high-quality individuals’ probability to solve the low efficiency caused by the loss of high-quality individuals in the dispersal process. This research also proposed SCA to improve the population generation and further improve the search efficiency of the algorithm. The influence of each improved part on the algorithm’s performance was quantitatively analyzed by multiple test functions and compared with many other improved algorithms for bacterial foraging. The results show that the improved Bacterial Foraging Algorithm proposed has the best results. This study applied the improved BFO to a microgrid economic scheduling model considering multiple optimization objectives to prove the excellent performance. MC is used to reduce the scene by aiming at the uncertainty processing of scenery force in the model. Finally, the experimental results prove that the method proposed can effectively reduce the operating cost of the microgrid, improve economic benefits, reduce environmental pollution, and ensure the stability of power consumption for users. In the future, we can study the accommodation of microgrids and the dispatching of microgrids with multiple energy storage devices.

## Figures and Tables

**Figure 1 biomimetics-08-00150-f001:**
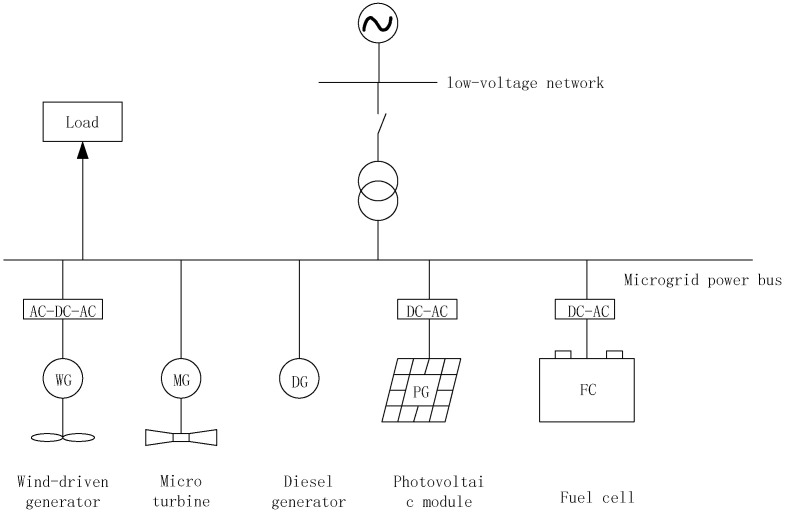
Schematic Diagram of Microgrid Structure.

**Figure 2 biomimetics-08-00150-f002:**
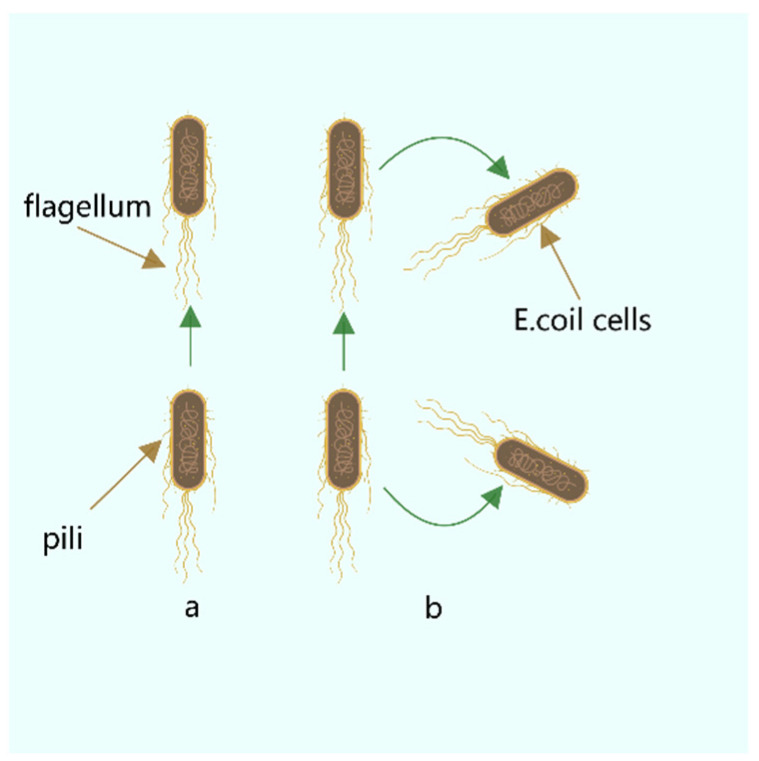
*E. coli* swimming flip chart.

**Figure 3 biomimetics-08-00150-f003:**
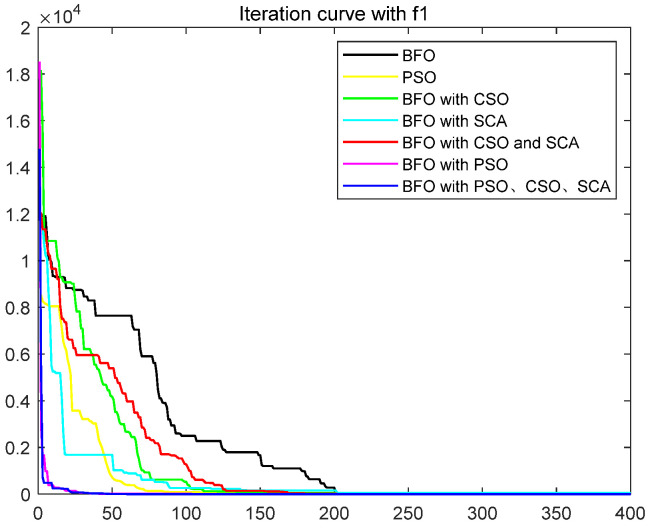
Convergence curve of f1.

**Figure 4 biomimetics-08-00150-f004:**
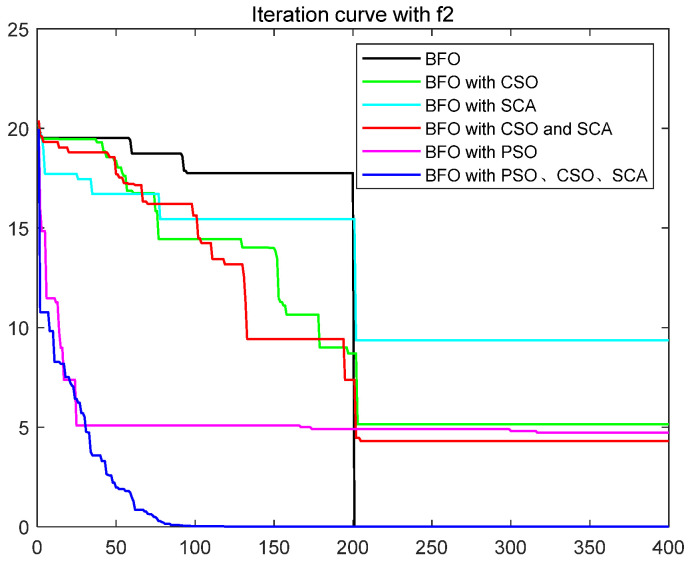
Convergence curve of f2.

**Figure 5 biomimetics-08-00150-f005:**
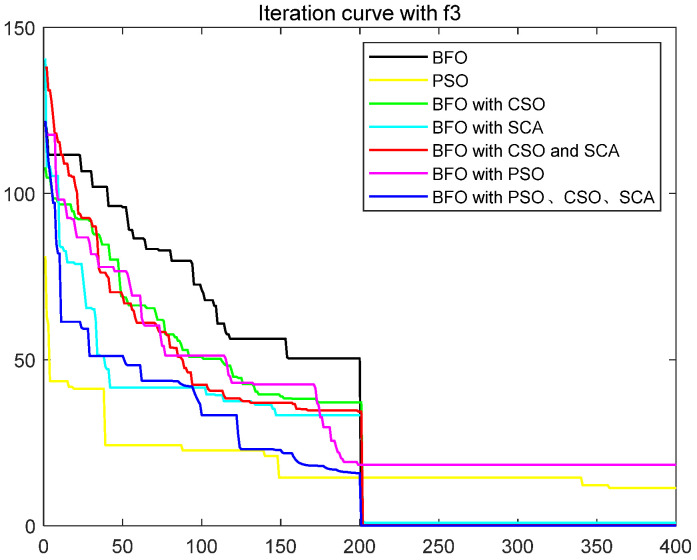
Convergence curve of f3.

**Figure 6 biomimetics-08-00150-f006:**
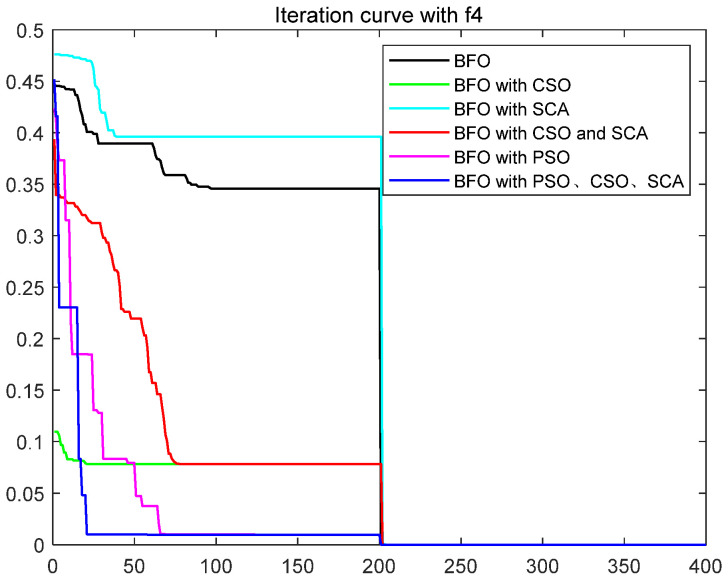
Convergence curve of f4.

**Figure 7 biomimetics-08-00150-f007:**
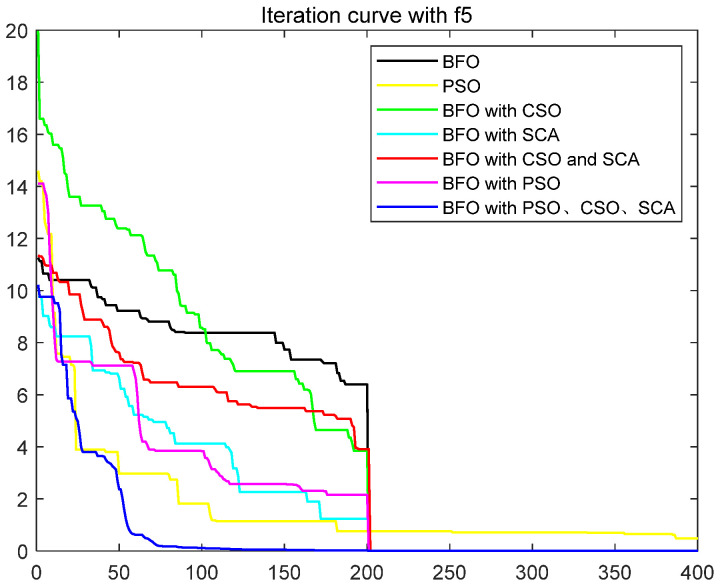
Convergence curve of f5.

**Figure 8 biomimetics-08-00150-f008:**
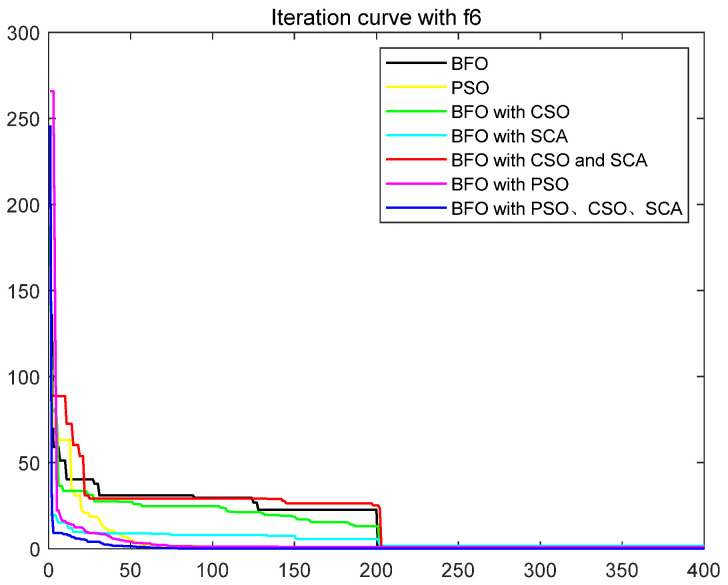
Convergence curve of f6.

**Figure 9 biomimetics-08-00150-f009:**
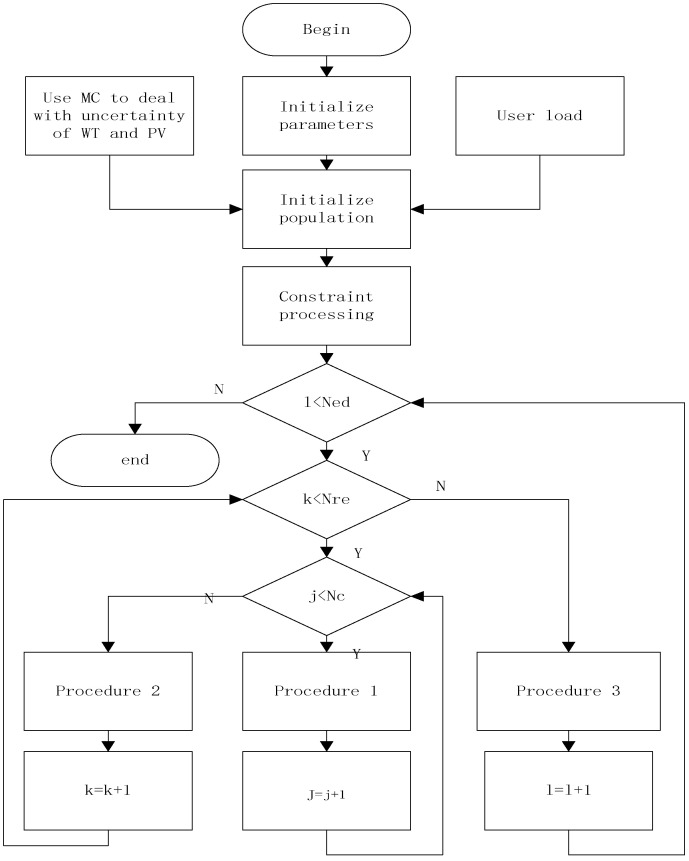
Flowchart of solving microgrid dispatching by comprehensively improved Bacterial Foraging Optimization.

**Figure 10 biomimetics-08-00150-f010:**
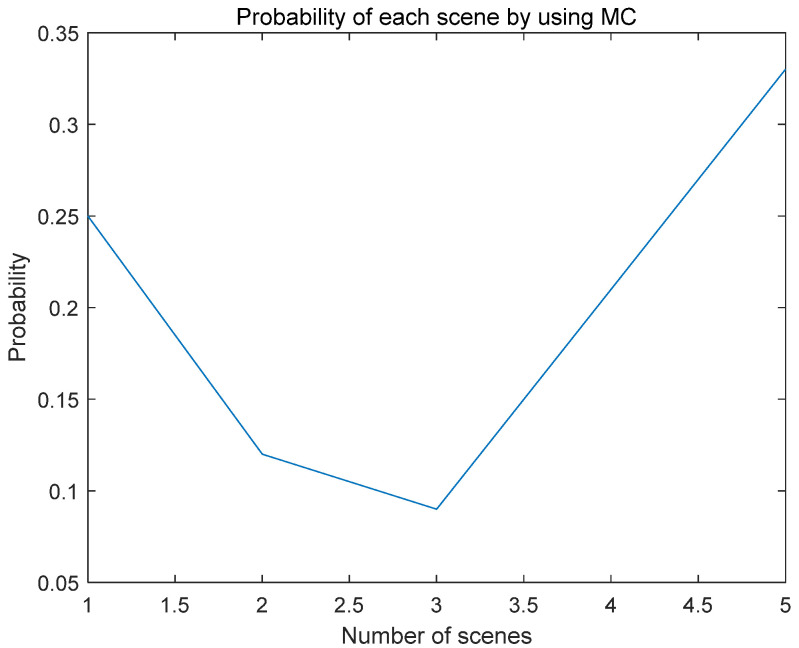
Reduced Scenario Probabilities.

**Figure 11 biomimetics-08-00150-f011:**
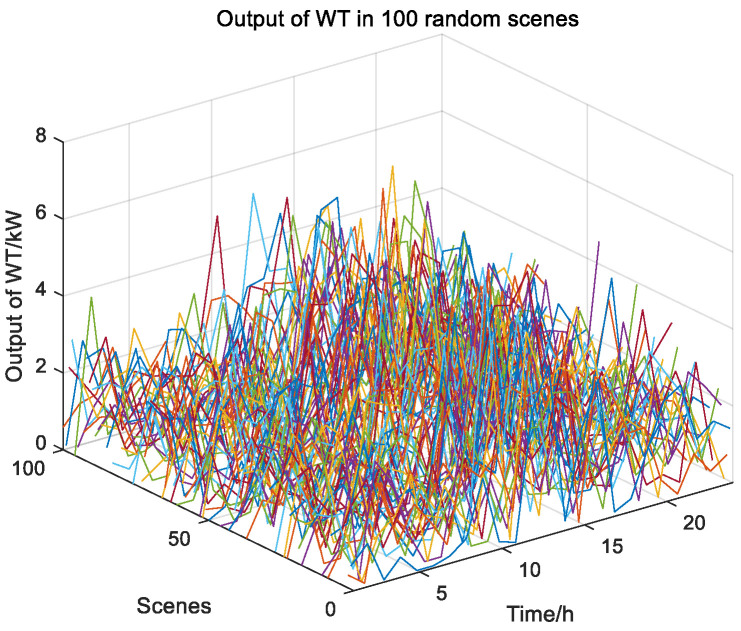
100 random scenarios of wind power output.

**Figure 12 biomimetics-08-00150-f012:**
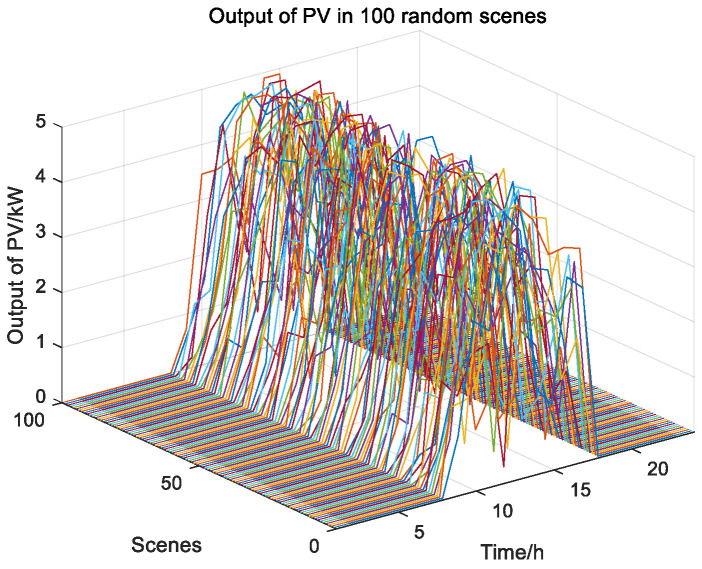
100 random scenarios of photovoltaic output.

**Figure 13 biomimetics-08-00150-f013:**
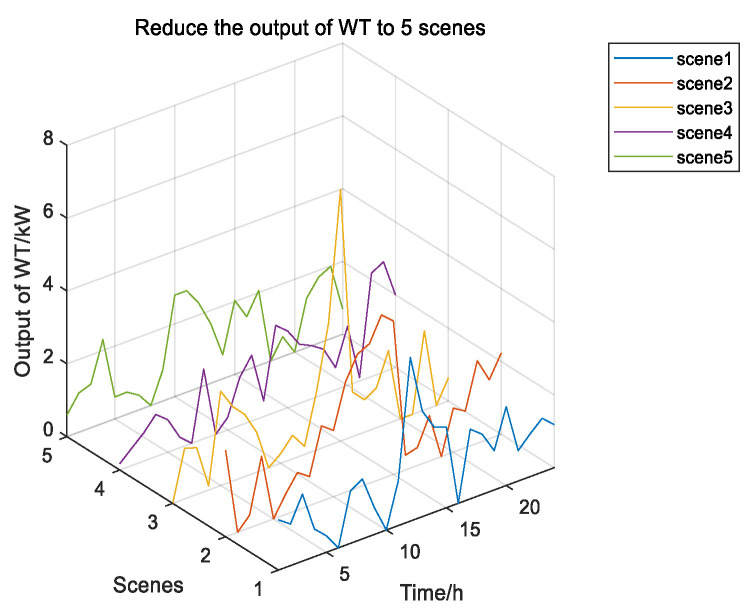
Wind power output reduced to 5 scenarios.

**Figure 14 biomimetics-08-00150-f014:**
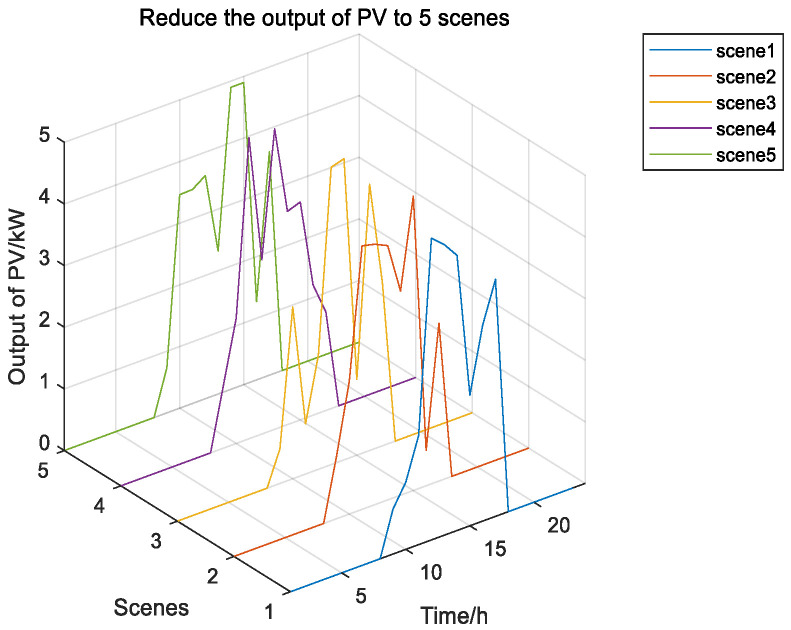
Photovoltaic output reduced to 5.

**Figure 15 biomimetics-08-00150-f015:**
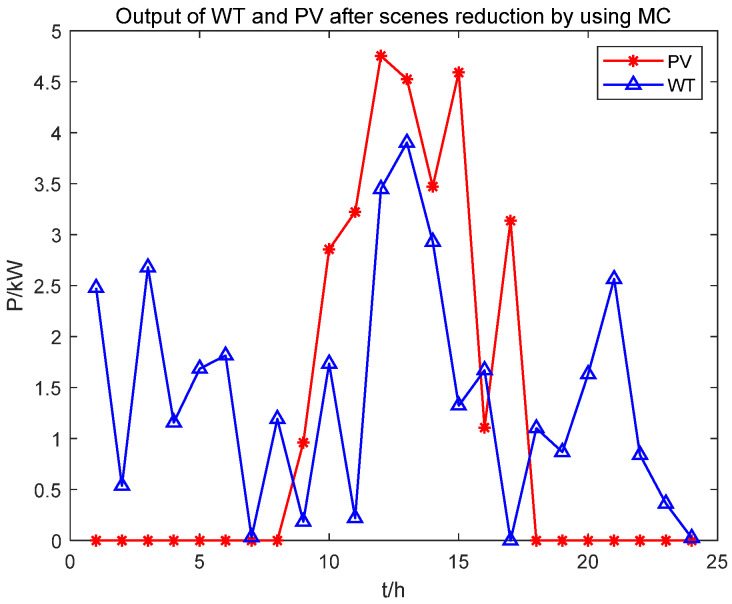
Wind power photovoltaic output after scenario reduction using MC.

**Figure 16 biomimetics-08-00150-f016:**
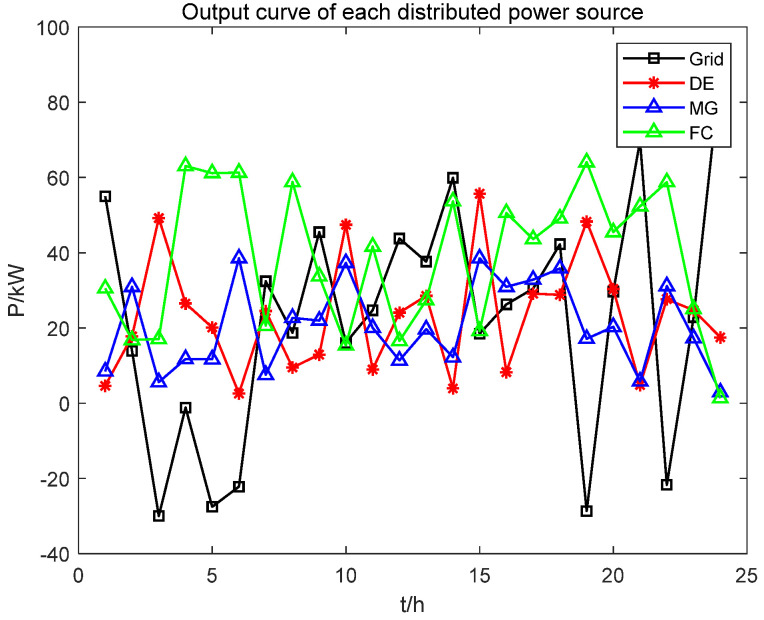
Scheduling output results of microgrid and distributed power sources.

**Figure 17 biomimetics-08-00150-f017:**
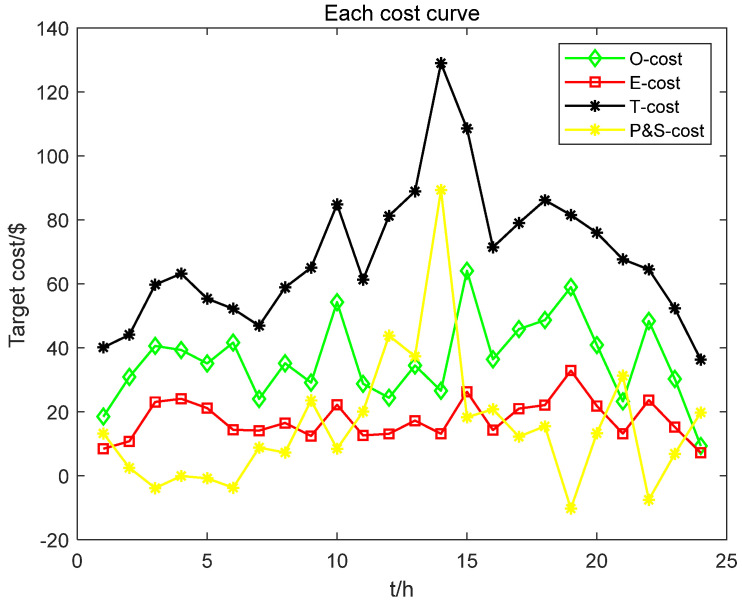
Each target cost result.

**Table 1 biomimetics-08-00150-t001:** Compare test results.

Function	Equation	Range	Optimization Technique	Best	Mean	std
Sphere	$f_{1}=\sum_{i=1}^{n}x_{i}^{2}$	[−100–100]	BFO	0.4159	0.5702	0.1178
			PSO	0.2074	0.7861	0.5161
			BFO with CSO	0.0355	0.0764	0.0196
			BFO with SCA	0.3719	0.6327	0.1740
			BFO with CSO and SCA	0.0270	0.0803	0.0178
			BFO with PSO [34]	0.0846	0.1316	0.0283
			BFO with PSO, CSO and SCA	3.82 × 10^−17^	4.79 × 10^−9^	1.34 × 10^−8^
			HR-EBFA [41]		2.69 × 10^−4^	2.29 × 10^−4^
			RL-BFA [42]		1.14 × 10^−2^	3.99 × 10^−3^
Ackley	$f_{2}=-20exp\left(-0.2\sqrt{\frac{1}{n}\sum_{i=1}^{n}x_{i}^{2}}\right)-exp\left(\frac{1}{n}\sum_{i=1}^{n}\cos\left(2\pi x_{i}\right)\right)+20+e$	[−32–32]	BFO	2.0337	2.3381	0.1838
			BFO with CSO	0.4498	0.6764	0.1175
			BFO with SCA	1.7569	2.2496	0.2334
			BFO with CSO and SCA	0.4263	0.7015	0.1197
			BFO with PSO [34]	0.7634	1.0645	0.1507
			BFO with PSO, CSO and SCA	3.36 × 10^−7^	0.0010	0.0017
			BFOED [43]	8.70 × 10^−5^	0.7134	0.2977
			BFOSA [43]	4.55 × 10^−3^	1.7133	0.3189
			MQBFA [44]		1.2485	
Rastrigin	$f_{3}=\sum_{i=1}^{n}\left[x_{i}^{2}-10\cos\left(2\pi x_{i}\right)+10\right]$	[−5.1–5.1]	BFO	18.5898	42.5381	8.1756
			PSO	7.7445	14.5918	4.2319
			BFO with CSO	8.4729	13.4023	2.3003
			BFO with SCA	24.6527	43.4678	7.8781
			BFO with CSO and SCA	4.0887	13.4822	3.3111
			BFO with PSO [34]	16.6043	26.8080	4.6352
			BFO with PSO, CSO and SCA	1.39 × 10^−12^	1.49 × 10^−7^	2.99 × 10^−7^
			HR-EBFA [41]		8.13 × 10^−4^	1.09 × 10^−3^
			RL-BFA [42]		1.9000	0.3140
			MQBFO [44]		25.6570	
Schaffer	$f_{4}=0.5+\frac{\sin^{2}\left(x_{1}^{2}-x_{2}^{2}\right)-0.5}{{\left[1+0.001\left(x_{1}^{2}-x_{2}^{2}\right)\right]}^{2}}$	[−100–100]	BFO	7.65 × 10^−8^	1.50 × 10^−6^	1.75 × 10^−6^
			BFO with CSO	1.83 × 10^−9^	4.88 × 10^−8^	4.53 × 10^−8^
			BFO with SCA	2.38 × 10^−8^	7.74 × 10^−7^	9.23 × 10^−7^
			BFO with CSO and SCA	4.99 × 10^−11^	9.48 × 10^−8^	7.95 × 10^−8^
			BFO with PSO [34]	1.72 × 10^−7^	9.48 × 10^−6^	1.64 × 10^−5^
			BFO with PSO, CSO and SCA	5.55 × 10^−17^	1.05 × 10^−10^	3.5 × 10^−10^
			HR-EBFA [41]		4.94 × 10^−3^	2.70 × 10^−3^
			RL-BFA [42]		2.69 × 10^−2^	3.82 × 10^−3^
Alpine	$f_{5}=\sum_{i=1}^{n}\left|x_{i}\sin x_{i}+0.1x_{i}\right|$	[−10–10]	BFO	0.1789	0.5285	0.1529
			PSO	0.0158	0.1231	0.1286
			BFO with CSO	0.0291	0.0671	0.0179
			BFO with SCA	0.2253	0.5181	0.1338
			BFO with CSO and SCA	0.0282	0.0599	0.0142
			BFO with PSO [34]	0.0582	0.1410	0.0492
			BFO with PSO, CSO and SCA	2.29 × 10^−8^	2.90 × 10^−4^	6.67 × 10^−4^
Schwefel	$f_{6}=\sum_{i=1}^{n}\left|x_{i}\right|+\prod_{i=1}^{n}\left|x_{i}\right|$	[−10–10]	BFO	1.0291	2.0283	0.3183
			PSO	0.2470	0.4224	0.0944
			BFO with CSO	0.4448	0.6637	0.1050
			BFO with SCA	1.2810	1.9379	0.2794
			BFO with CSO and SCA	0.5394	0.7073	0.0835
			BFO with PSO [34]	0.7039	0.9021	0.1191
			BFO with PSO, CSO and SCA	1.93 × 10^−7^	0.0016	0.0046

**Table 2 biomimetics-08-00150-t002:** Basic parameters of controllable distributed power supply.

Controllable Micro Power Type	Life Expectancy/Year	Power Lower Limit/KW	Power Upper Limit/KW
DE	10	0	60
FC	10	0	40
MT	10	0	65
grid		−30	200

**Table 3 biomimetics-08-00150-t003:** Typical daily load and real-time electricity price.

Time Period/h	Load/KW	Electricity Price/(Yuan/(KW·h))	Time Period/h	Load/KW	Electricity Price/(Yuan/(KW·h))
00:00–01:00	101.049	0.2400	12:00–13:00	121.629	0.9900
01:00–02:00	79.991	0.1770	13:00–14:00	136.151	1.4900
02:00–03:00	41.862	0.1301	14:00–15:00	137.752	0.9900
03:00–04:00	101.312	0.0969	15:00–16:00	118.824	0.7900
04:00–05:00	67.139	0.0300	16:00–17:00	139.221	0.4000
05:00–06:00	82.000	0.1701	17:00–18:00	157.158	0.3647
06:00–07:00	85.085	0.2710	18:00–19:00	101.689	0.3590
07:00–08:00	110.875	0.3864	19:00–20:00	127.400	0.4130
08:00–09:00	115.249	0.5169	20:00–21:00	135.312	0.4448
09:00–10:00	120.687	0.5260	21:00–22:00	96.692	0.3480
10:00–11:00	98.786	0.8100	22:00–23:00	90.243	0.3000
11:00–12:00	13.944	1.0000	23:00–24:00	109.587	0.2250

**Table 4 biomimetics-08-00150-t004:** Pollutant discharge and cost coefficient.

Types of Polluting Gases	Treatment Cost (Yuan/kg)	Controllable Power Supply Pollution Gas Emission Coefficient (g/(KW·h))		
		DE	MT	FC
$NO_{x}$	26.46	3.74	1.82	0.01
$SO_{2}$	6.237	8.79	2.28	0.003
$CO_{2}$	0.21	1142.9	724.6	20.4

**Table 5 biomimetics-08-00150-t005:** Detailed data of microgrid and the output of DG.

Time (h)	Grid (KW)	DE (KW)	MT (KW)	FC (KW)	PV-WT (KW)
1	54.9847	4.6157	30.5293	8.4419	2.4774
2	13.9208	17.7461	16.7995	30.9069	0.5377
3	−30.0000	49.2007	17.0434	5.5785	2.6790
4	−1.1387	26.5018	63.0595	11.7369	1.1584
5	−27.5462	20.1368	61.1810	11.6811	1.6862
6	−22.2145	2.5391	61.3192	38.5415	1.8147
7	32.4267	24.4806	20.6001	7.5464	0.0312
8	18.7005	9.4939	58.8034	22.6857	1.1915
9	45.4677	12.9296	33.7325	21.9738	1.1454
10	16.0335	47.4314	15.3470	37.2843	4.5908
11	24.6919	8.9679	41.6544	20.0269	3.4449
12	43.7713	24.0744	16.5186	11.3769	8.2029
13	37.6223	28.5550	27.3770	19.6472	8.4275
14	59.9395	3.9189	53.6826	12.2077	6.4023
15	18.4272	55.6835	19.2147	38.5125	5.9142
16	26.2810	8.2503	50.6074	30.9076	2.7776
17	30.4756	29.1353	43.6213	32.8522	3.1366
18	42.2725	28.8073	49.1521	35.8254	1.1007
19	−28.6802	48.2836	64.0777	17.1399	0.8680
20	29.6620	30.3528	45.4651	20.2881	1.6321
21	69.7417	4.8349	52.3628	5.8072	2.5654
22	−21.7034	27.6313	58.7886	31.1348	0.8406
23	22.8784	24.7073	25.0228	17.2717	0.3629

## Data Availability

The original contributions presented in the study are included in the article, further inquiries can be directed to the corresponding author.

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
