# Peer review of "Research on Economic Optimal Dispatching of Microgrid Based on an Improved Bacteria Foraging Optimization"

_biomimetics, 2023, doi:10.3390/biomimetics8020150_

Round 1

Reviewer 1 Report

Major revision is required. And my specific comments are shown below:

1-               The research paper should be written in the perspective of the third person. Words such as ‘we’ etc. needs to be avoided. Such as:  " we used the dynamic probability....", " We propose the dy-500 namic dispersal equation to...", "We also reduce the step size …", etc.

2-               please separate the section "study presents some improvements in Bacterial 81 Foraging Optimization (BFO).... " by new subsection called "Contributions Section"

3-               Authors must develop original figures with high resolution for proper rendering of the final published paper and should not distort on the zooming. such as: "Fig.1 Schematic Diagram of Microgrid Structure ", " Fig.11 100 random scenarios of wind power output ",  "Fig.13 Wind power output reduced to 5 scenarios  ", etc.

4-               please delete the "new " words from the paper such as "As new energy generation forms...", because your paper did not use a new algorithm

5-               There are many combined words, such as: ThingSpeakIoT, MathWorksThingSpeak,. etc..

6-               No comparison with other previous works !!, the results should be compared with the most recent methods in the literature

7-               .The authors may enrich their references with the latest and related work further, such as below, and more

·        A. Khalid, N. Javaid, A. Mateen, B. Khalid, Z. A. Khan and U. Qasim, "Demand Side Management Using Hybrid Bacterial Foraging and Genetic Algorithm Optimization Techniques," 2016 10th International Conference on Complex, Intelligent, and Software Intensive Systems (CISIS), Fukuoka, 2016, pp. 494-502, doi: 10.1109/CISIS.2016.128.

·        Bilal Naji Alhasnawi, Basil H. Jasim, Pierluigi Siano, Hassan Haes Alhelou, and Amer Al-Hinai, "A Novel Solution for Day-Ahead Scheduling Problems Using the IoT-Based Bald Eagle Search Optimization Algorithm ", Inventions 2022, 7(3), 48; https://doi.org/10.3390/inventions7030048

·        Hernández-Ocaña, B.; Hernández-Torruco, J.; Chávez-Bosquez, O.; Calva-Yáñez, M.B.; Portilla-Flores, E.A. Bacterial Foraging-Based Algorithm for Optimizing the Power Generation of an Isolated Microgrid. Appl. Sci. 2019, 9, 1261. https://doi.org/10.3390/app9061261

·        Alhasnawi, B.N.; Jasim, B.H.; Mansoor, R.; Alhasnawi, A.N.; Rahman, Z.-A.S.A.; Haes Alhelou, H.; Guerrero, J.M.; Dakhil, A.M.; Siano, P. A new Internet of Things based optimization scheme of residential demand side management system. IET Re-new. Power Gener. 2022, 1–15. https://doi.org/10.1049/rpg2.12466 .

Author Response

Re: Manuscript ID: biomimetics-2237041 and Title: Research on Economic Optimal Dispatching of Microgrid Based on An Improved Bacteria Foraging Optimization.

Thank you for the reviewers’ comments concerning our manuscript entitled “Research on Economic Optimal Dispatching of Microgrid Based on An Improved Bacteria Foraging Optimization” (biomimetics-2237041). Those comments are valuable and very helpful. We have read through the comments carefully and have made corrections. We uploaded the file of the revised manuscript based on the instructions provided in your letter. Revisions in the text are shown using the red highlight for addition. The responses to the reviewer's comments presented following.

As a master’s student, this is my first time submitting a paper to biomimetics. I am very grateful to thank you for allowing us to resubmit a revised copy of the manuscript and we highly appreciate your time and consideration.

Sincerely.

Lv Yang .

Q1:The research paper should be written in the perspective of the third person. Words such as ‘we’ etc. needs to be avoided. Such as:  " we used the dynamic probability....", " We propose the dy-500 namic dispersal equation to...", "We also reduce the step size …", etc.

Response: We modify the sentences with the perspective of the third person.

Q2. please separate the section "study presents some improvements in Bacterial 81 Foraging Optimization (BFO).... " by new subsection called "Contributions Section"

Response: line 110

Q3.  Authors must develop original figures with high resolution for proper rendering of the final published paper and should not distort on the zooming. such as: "Fig.1 Schematic Diagram of Microgrid Structure ", " Fig.11 100 random scenarios of wind power output ",  "Fig.13 Wind power output reduced to 5 scenarios  ", etc.

Response: line129,473,480

Q4. please delete the "new " words from the paper such as "As new energy generation forms...", because your paper did not use a new algorithm.

Response: line 136-137

Q5. There are many combined words, such as: ThingSpeakIoT, MathWorksThingSpeak, etc.

Response: We modify the sentences according to the reviewer.

Q6. No comparison with other previous works !!, the results should be compared with the most recent methods in the literature.

Response: line 426

Q7.The authors may enrich their references with the latest and related work further, such as below, and more

  • A. Khalid, N. Javaid, A. Mateen, B. Khalid, Z. A. Khan and U. Qasim, "Demand Side Management Using Hybrid Bacterial Foraging and Genetic Algorithm Optimization Techniques," 2016 10th International Conference on Complex, Intelligent, and Software Intensive Systems (CISIS), Fukuoka, 2016, pp. 494-502, doi: 10.1109/CISIS.2016.128.
  • Bilal Naji Alhasnawi, Basil H. Jasim, Pierluigi Siano, Hassan Haes Alhelou, and Amer Al-Hinai, "A Novel Solution for Day-Ahead Scheduling Problems Using the IoT-Based Bald Eagle Search Optimization Algorithm ", Inventions 2022, 7(3), 48; https://doi.org/10.3390/inventions7030048
  • Hernández-Ocaña, B.; Hernández-Torruco, J.; Chávez-Bosquez, O.; Calva-Yáñez, M.B.; Portil

Response: We reinserted references according to the reviewer.

Reviewer 2 Report

The proposed paper presents an enhanced bacterial foraging algorithm to tackle the scheduling problem of microgrid. The improvements aim to enhance the algorithm's performance, and the comparative analysis illustrates the specific impact of each improved part. The effectiveness of the improved algorithm is demonstrated by applying it to microgrid dispatching with multiple distributed power sources. The paper should be refined in the following ways:

(1) The improvement of the scheduling results by the proposed algorithm should be presented using percentage values in the summary.

(2) The introduction should provide a detailed account of the development and improvement of swarm intelligence algorithms in the field of microgrid dispatching.

(3) The impact of changing r4 value in formulas (26) and (27) on the overall formula should be explained in more detail.

(4) The introduction should be expanded to include a detailed discussion of the processing of wind and light data, which is a hot research topic in the field.

(5) A list of terms and abbreviations should be added to enhance the paper's clarity and readability.

DETAILED COMMENTS FOR AUTHORS:

Typesetting of the paper is without due care.

All the titles of figures, tables, and equations should end with “.”.

Throughout the paper, some spaces after the reference are missing.

E.g. P4, L152, Ref[20]proposed-> Ref[20] proposed.

Revise the poor formatted equations (4), (5), and (17).

P3,L109. In Eq (1), variables Pw and Pn are not defined. In Eq (2), Pv is not defined.

P7,L273. In Eq (18)-(27), ‘*’ is used for multiplication, which is inconsistent with other equations.

P8,L287. For all the procedures, define the input and output of the pseudocode. Define all the variables in the procedure as well. Do not use “equation(*)” in the pseudocode. Instead, either define a function for the equation or copy the equation. For example, on P8, Line 3 of the pseudocode, use “C(x)” rather than “equation(18)” and define the parameter x of C().

In Figures 3-8, the name and unit of parameters of coordinates are missing. In Table 5, unit of the data is missing.

Author Response

Re: Manuscript ID: biomimetics-2237041 and Title: Research on Economic Optimal Dispatching of Microgrid Based on An Improved Bacteria Foraging Optimization.

Thank you for the reviewers’ comments concerning our manuscript entitled “Research on Economic Optimal Dispatching of Microgrid Based on An Improved Bacteria Foraging Optimization” (biomimetics-2237041). Those comments are valuable and very helpful. We have read through the comments carefully and have made corrections. We uploaded the file of the revised manuscript based on the instructions provided in your letter. Revisions in the text are shown using the green highlight for addition and we respond to the reviewer with the line marked. The responses to the reviewer's comments presented following.

As a master’s student, this is my first time submitting a paper to biomimetics. I am very grateful to thank you for allowing us to resubmit a revised copy of the manuscript and we highly appreciate your time and consideration.

Sincerely.

Lv Yang .

Q1: improvement of the scheduling results by the proposed algorithm should be presented using percentage values in the summary.

Response: Line19-21

Q2:The introduction should provide a detailed account of the development and improvement of swarm intelligence algorithms in the field of microgrid dispatching.

Response: Line104-108

Q3: The impact of changing r4 value in formulas (26) and (27) on the overall formula should be explained in more detail.

Response: Line 398-402

Q4:The introduction should be expanded to include a detailed discussion of the processing of wind and light data, which is a hot research topic in the field.

Response: Lin42-48

Q5: A list of terms and abbreviations should be added to enhance the paper's clarity and readability.

Response: Line24-26

Q6: DETAILED COMMENTS FOR AUTHORS:

Response: We modify the sentences according to the reviewer.

Reviewer 3 Report

Some suggestions for improving the paper, both in content and in presentation:

- a short insight in BFO would be useful for the readers;

- equations in line with text could be rewritten, they appear on the upper part of the rows (such as in rows 314, 332, 340, 341,...);

- unify the presentation of the procedures (for example, Procedure 1 and Procedure 2 appear very different);

- Figures 11, 12 must be reorganized, 13 and 14 appear too small;

- Some indications regarding the computers the experiment has been organized on could be useful. How do the BFO improvements look measured in terms of computing time and memory?

After reference 38 you have now reference no 30...

Author Response

Re: Manuscript ID: biomimetics-2237041 and Title: Research on Economic Optimal Dispatching of Microgrid Based on An Improved Bacteria Foraging Optimization.

Thank you for the reviewers’ comments concerning our manuscript entitled “Research on Economic Optimal Dispatching of Microgrid Based on An Improved Bacteria Foraging Optimization” (biomimetics-2237041). Those comments are valuable and very helpful. We have read through the comments carefully and have made corrections. We uploaded the file of the revised manuscript based on the instructions provided in your letter. Revisions in the text are shown using the blue highlight for addition and we respond to the reviewer with the row marked. The responses to the reviewer's comments presented following.

As a master’s student, this is my first time submitting a paper to biomimetics. I am very grateful to thank you for allowing us to resubmit a revised copy of the manuscript and we highly appreciate your time and consideration.

Sincerely.

Lv Yang .

Q1- a short insight in BFO would be useful for the readers;

Response: row 271-273

Q2- equations in line with text could be rewritten, they appear on the upper part of the rows (such as in rows 314, 332, 340, 341,...);

Response: row233,280,297,313,331,369,378,478

Q3- unify the presentation of the procedures (for example, Procedure 1 and Procedure 2 appear very different);

Response: row 324-325,364-365,411-412

Q4- Figures 11, 12 must be reorganized, 13 and 14 appear too small;

Response: row 477,479

Q5- Some indications regarding the computers the experiment has been organized on could be useful. How do the BFO improvements look measured in terms of computing time and memory?

Response: row 532-541

Q6-After reference 38 you have now reference no 30..

Response: We reinserted references

Round 2

Reviewer 1 Report

no comments